

# Optimization of U-shaped pure transformer medical image segmentation network

Yongping Dan, Weishou Jin, Zhida Wang and Changhao Sun

School of Electronic and Information, Zhongyuan University of Technology, Zhengzhou, Henan, China

## ABSTRACT

In recent years, neural networks have made pioneering achievements in the field of medical imaging. In particular, deep neural networks based on U-shaped structures are widely used in different medical image segmentation tasks. In order to improve the early diagnosis and clinical decision-making system of lung diseases, it has become a key step to use the neural network for lung segmentation to assist in positioning and observing the shape. There is still the problem of low precision. For the sake of achieving better segmentation accuracy, an optimized pure Transformer U-shaped segmentation is proposed in this article. The optimization segmentation network adopts the method of adding skip connections and performing special splicing processing, which reduces the information loss in the encoding process and increases the information in the decoding process, so as to achieve the purpose of improving the segmentation accuracy. The final experiment shows that our improved network achieves 97.86% accuracy in segmentation of the "Chest Xray Masks and Labels" dataset, which is better than the full convolutional network or the combination of Transformer and convolution.

# INTRODUCTION

With the development of deep learning, computer vision technology has made immense splash in the field of medical image analysis. Medical image segmentation has become an important branch of medical image analysis (*Chen et al., 2021*; *Li et al., 2022*, *2023*; *Yue et al., 2022*). Stable and highly accurate medical image segmentation can greatly improve the clinical speed and diagnostic accuracy of doctors.

Technological developments have led to an increased focus on more comprehensive anatomical models (*Simpson et al., 2019*), which has led to the development of models for organ analysis. In the context of organ analysis, the brain and abdomen have emerged as the most popular areas of medical image analysis. Rapid advances in imaging techniques and deep learning techniques have resulted in numerous datasets for different applications in different organs. These data sets can be used to train a dedicated medical segmentation network model that can segment important organs, tissues, or lesions in the image and extract the segmented object features. Anatomical models can be constrained and labeled with contextual information from stable abdominal structures (*e.g.*, liver, spleen, kidneys,

Corresponding author
Yongping Dan, 420076822@qq.com

stomach, pleural effusion) as well as the pelvic cavity (colon, prostate) (*Heller et al., 2021*; *Ma et al., 2022*, *2021*). In addition, there are many studies on human tumors, such as brain tumors, abdominal tumors, head and neck tumors, breast tumors, *etc.* (*Heller et al., 2022*; *Bilic et al., 2023*; *Clark et al., 2013*). The latest ones, such as *Yuan et al. (2023)*, have an average segmentation accuracy of 77.97% and 69.04% respectively in pancreatic tumors and liver tumors. Accurate segmentation is crucial for clinical applications, including disease diagnosis, treatment planning, and disease progression detection.

At the present stage, medical image segmentation technology mainly applies the U-shaped structure of the full convolutional neural network (FCNN) (*Long, Shelhamer & Darrell, 2015*). The classical U-shaped structure network consists of a symmetric encoder-decoder with skip connections, also known as U-Net (*Guan et al., 2020*; *He et al., 2016*). In the encoder, numerous convolutional and downsampling layer combinations are used to extract deep features with large sensory fields at different scales. Then, the decoder up-samples the extracted deep features to the resolution of the initial input image and fuses them with the different scale features in the encoder introduced by the skip connections, achieving the goal of improving the prediction accuracy by reducing the information loss in the downsampling process. Such an efficient and simple structural design has enabled U-Net to achieve great success in the field of medical images. Continuing this design idea, a series of algorithms such as Res-Unet (*Xiao et al., 2018*), R2U-Net (*Alom et al., 2018*), U-Net++ (*Zhou et al., 2020*), and UNet3+ (*Huang et al., 2020*) have been developed for 2D medical image segmentation tasks (*Litjens et al., 2017*). Numerous convolutional neural network (CNN) based methods have demonstrated that CNNs are highly capable at segmentation tasks.

Currently, CNN-based segmentation methods (*Girshick et al., 2015*; *Bo et al., 2017*; *Lee et al., 2017*) have achieved excellent results in medical image tasks, but they still cannot fully satisfy the demand for high accuracy in medical image segmentation tasks. In addition, the limitations of convolutional operations make it difficult for the CNN approach to learn explicit global and long-range semantic information. As Transformer has become the dominant network in the field of natural language processing (NLP), researchers have tried to apply it to semantic segmentation tasks, and the local operations of convolution and the global operations of Transformer operations well complement each other (*Vaswani et al., 2017*). U-shaped segmentation networks combining CNN and Transformer, such as TransUNet (*Chen et al., 2021*), emerged to exploit the advantages of each for hybrid coding, where the powerful global capability of Transformer and the ability of CNN to focus on image details at low resolution to overcome the problem of long-range contextual interactions improved the segmentation accuracy. In *Liu et al. (2021)*, a new vision transformer called Swin-Transformer is proposed as a generic backbone to perform image recognition tasks. Inspired by Swin Transformer, researchers then proposed Swin-Unet (*Cao et al., 2021*), which replaced the original CNN-based composition of encoders and decoders with the Swin Transformer block to obtain a U-shaped segmentation network with pure Transformer.

Swin-Unet has high precision for medical segmentation tasks. Although skipping connections is used to reduce the loss of spatial information in the downsampling process,

a large amount of information loss will still affect segmentation accuracy. In order to deal with this problem, an improved Swin-Unet is proposed in this article. The improved U-shaped network consists of encoders, decoders, and skip connections, as well as our addition of multi-scale skip connections and special splicing modules. By adding multi-scale skip connections, features from different scales of the encoding process and features from the sampling process on the decoder are introduced for special splicing and fusion, thus obtaining feature maps that aggregate more information and perform segmentation prediction. Experiments conducted on the lung dataset show improved network segmentation prediction accuracy. Specifically, our contributions are summarized as: (1) the addition of asymmetric skip connections in the U-shaped network, which capture more spatial information. (2) The creation of a new splicing and fusion module that is able to fuse feature information from adjacent scales in the encoder and upsample features in the decoder thus achieves the purpose of increasing the prediction accuracy of segmentation.

# RELATED WORK

## CNN-based model

Early medical image segmentation was mainly based on traditional machine learning techniques (*Mcinerney & Terzopoulos, 1996*; *Boykov & Funka-Lea, 2006*; *Staal et al., 2004*) such as edge detection-based segmentation algorithms and aggregation-based segmentation algorithms. With the continuous development of CNN, U-Net, based on the FCN network (*Long, Shelhamer & Darrell, 2015*), was proposed to achieve a big leap in the overall accuracy of medical image segmentation. Due to the concise and efficient U-shaped structure, various U-based methods have been generated, such as U-Net++ and UNet3+. And it has been extended from 2D segmentation to 3D segmentation, such as in 3D-Unet (*Kafali et al., 2021*), Dense-U-Net (*Wu et al., 2021*), and KiU-Net (*Valanarasu et al., 2020*). At this stage, CNN-based methods have achieved great success in the field of medical image segmentation.

## Transformer to complement CNNs

U-shaped structures have become the *de facto* standard in various medical image segmentation tasks, and researchers have introduced attention mechanisms into CNN networks in order to improve network performance. In *Chen et al. (2021)*, the self-attention mechanism is integrated into the U-shaped structure for medical image segmentation. The researchers combined CNN and Transformer, where the Transformer encodes the feature maps from CNN as the input sequence for extracting the context, and the encoder still uses the convolutional network to upsample the encoded features. The combination of the two enhances finer details and improves segmentation accuracy. However, these are still CNN-based methods.

## Vision transformers

The Transformer was proposed in *Vaswani et al. (2017)* to be applied to machine translation tasks (*Nie et al., 2017*). The powerful global modeling capabilities of the

Transformer, together with its excellent transferability to downstream tasks under large-scale pre-training, have made it a great success in the fields of machine translation and natural language processing (NLP) (*Chen et al., 2018*). Driven by the great success of the Transformer, researchers have proposed a novel Vision Transformer (VIT) (*Dosovitskiy et al., 2022*) that interprets images as a series of patches and processes them with the standard Transformer encoder used in NLP, which has achieved surprising speed and accuracy in image detection and segmentation tasks. In contrast to CNN-based models, VIT has the disadvantage that it requires pre-training processing on large datasets. Recently, several works have been done on VIT to alleviate the difficulties in its training process. It is worth noting that an efficient vision transformer with hierarchy was proposed in *Ze et al. (2021)* as a new vision backbone, called Swin Transformer. Based on the hierarchy-shifted window approach, Swin Transformer has achieved excellent performance on various vision tasks. After some researchers built a U-shaped encoder-decoder segmentation network using Swin Transformer as a backbone but found that it had shortcomings, we tried to improve it and build a new medical semantic segmentation network with better performance.

## METHODS

### Overall architecture

The overall structure mentioned in this article is as shown in the Fig. 1. This design consists of encoder, decoder, and skip connections. The Swin Transformer is the basic unit block. For the encoder, the medical image is segmented into non-overlapping $4 \times 4$ patches (*Li et al., 2023*) of varying sizes by a patch splitting module. In addition, a linear embedding layer maps the raw-valued features to arbitrary dimensions. The mapped output patch vector generates a hierarchical feature representation through several Swin Transformer blocks and patch merging layers. In brief, the patch merging layer is applied to downsample and increase dimensions, and the Swin Transformer Block is responsible for learning feature representation. For the skip connections, inspired by U-Net++ (*Zhou et al., 2020*), the number of connections is increased on the basis of the original skip connection. The encoder is composed of the Swin Transformer, the patch expanding layer, and the patch Splicing layer. The extracted context information is multi-scale fused by the patch splicing module through skip connections to supplement the spatial information loss in the down-sampling process. The patch expanding layer is designed to sample and reduce dimensions to obtain a higher resolution feature map. In the last patch expanding layer, the feature map is recovered to the input image pixel size by quadruple upsampling. Finally, the obtained features are applied to the linear mapping layer to output pixel-level segmentation prediction. The role of each module is explained in detail below.

### Swin Transformer block

Compared with the traditional multi-head self-attention (MSA) module in the NLP network, the Swin Transformer block uses more advanced shifted window-based multi-head attention (W-MSA and SW-MSA) modules. Non-overlapping windows and cross-window connections are conducive to more effective modeling. As shown in Fig. 2, two

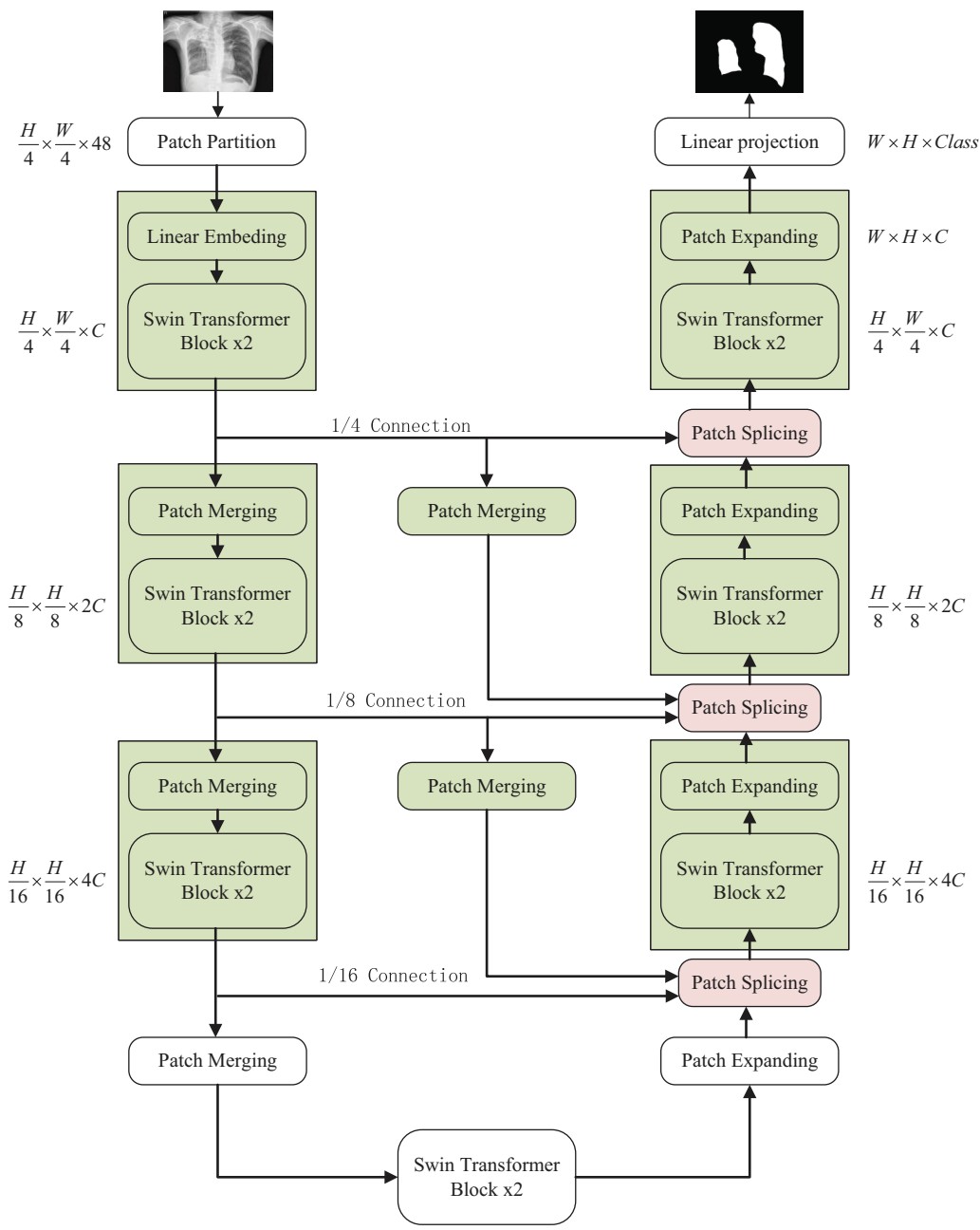

**Figure 1** **The overall structure of the optimized model: the left half is the encoder, the right half is the decoder, and the middle is composed of multiple skip connections.** Image credit: "Chest Xray Masks and Labels" dataset (https://www.kaggle.com/datasets/nikhilpandey360/chest-xray-masks-and-labels); License: CC0: Public Domain.     

consecutive Swin Transformer blocks are shown. Each Swin Transformer block consists of a multi-attention module based on a mobile window, a two-layer MLP with GELU nonlinear activation, and two LayerNorm (LN) layers that are normalized.

The two attention modules W-MSA and SW-MSA in the block use different window configurations, and based on this window mechanism, the consecutive Swin Transformer block can be represented as:

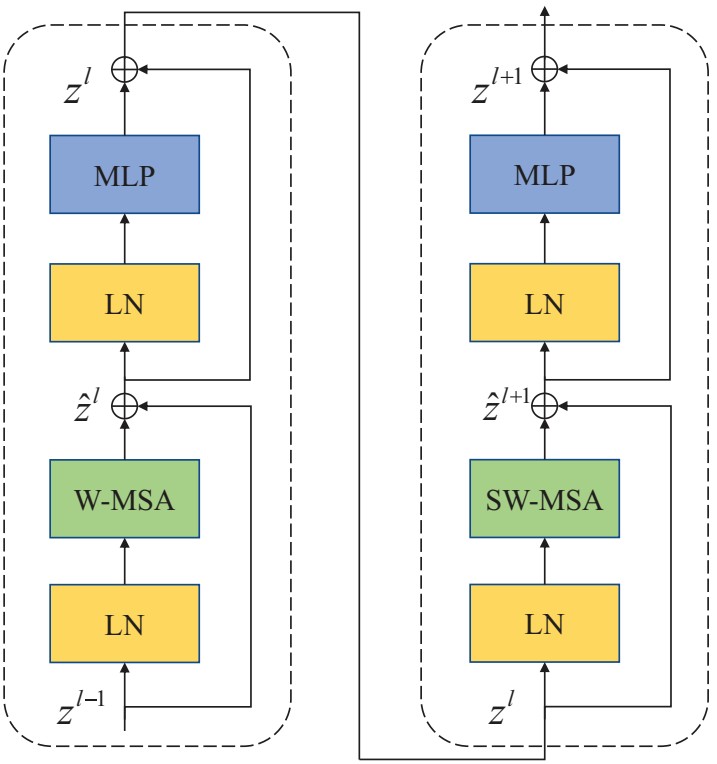

**Figure 2** Swin Transformer block (W-MSA is a multi-head self-attention module with conventional configuration, and SW-MSA is a multi-head self-attention module based on shifted window configuration).

$$\hat{z}^l = W - MSA\big(LN\big(z^{l-1}\big)\big) + z^{l-1} \tag{1}$$

$$z^l = MLP\Big(LN\big(\hat{z}^l\big)\Big) + \hat{z}^l \tag{2}$$

$$\hat{z}^{l+1} = SW - MSA\big(LN\big(z^l\big)\big) + z^l \tag{3}$$

$$z^{l+1} = MLP\Big(LN\big(\hat{z}^{l+1}\big)\Big) + \hat{z}^{l+1} \tag{4}$$

Similar to the traditional self-attention calculation method, where $\hat{z}^l$ and $z^l$ represent the output of the first W-MSA module and the MLP module, respectively.

$$Attention(Q, K, V) = SoftMax\bigg(\frac{QK^T}{\sqrt{d}} + B\bigg)V \tag{5}$$

where $Q, K, V \in \mathbb{R}^{M^2 \times d}$ respectively represents matrix query, matrix key and value. M represents the number of patches in a window and d represents the dimensionality of query and key. Since the relative positions of the axes are at [−M+1, M−1], Therefore the value of B comes from the bias matrix $\hat{B} \in \mathbb{R}^{(2M-1) \times (2M+1)}$.

## Encoder

In the encoder, the original image being partitioned and processed is mapped to C dimension, and then the data input with C dimension pixel size of $H/4 \times W/4$ tokens is

fed to two consecutive Swin Transformer blocks for feature learning with feature size and resolution kept constant before and after processing. At the same time, to produce the layered representation, each patch merging layer will perform 2 × down-sampling to reduce the number of tokens and increase the feature dimension to 2 × the original dimension. The above operation is repeated to obtain layered feature maps at different scales similar to those in convolutional networks.

**Patch merging layer**: To reduce the resolution and increase the dimensionality of the features, the input patches are decomposed into four parts and then merged together to achieve a two-fold downsampling operation and a four-fold increase in dimensionality. Since the dimension is increased to four times the original dimension, a linear layer is applied to unify the feature dimension to two times the original dimension.

## Decoder

Similar to the encoder, the decoder is also built based on the Swin Transformer block. To restore the feature map to the input image size and dimensions, a patch expanding layer is applied to upsample the extracted features, as opposed to the patch merging layer in the encoder. With the patch expanding layer operation, the feature map is reconstructed to a higher resolution feature map (2 × upsampling) and the feature dimension is reduced to half of the original dimension.

**Patch expanding layer**: In a patch expanding layer, first a linear layer increases the input feature dimension to twice the input dimension. Immediately afterwards, using rearrangement and image transformation operations, the feature resolution is expanded to twice the original input pixels and the feature dimension is reduced to one-half of the input dimension. With the above processing, the feature dimension becomes one-half of the initial dimension and the feature size is expanded to twice the original input pixels.

**Patch splicing layer**: The patch splicing layer is designed to fuse the multiscale features of the encoding process with the upsample features. This is shown in Fig. 3. In the first two patch splicing layers, the information ($X^1$ and $X^2$) of the two scales in the encoding process is concatenated, and the feature dimension is increased to twice the original input dimension. Subsequently, a linear layer is applied to reduce the dimensionality to the original input feature dimension. Then the same operation is performed with the upsampled feature information $X^3$ to obtain the fused output feature $Y$. The last patch splicing layer directly fuses the two sets of feature information using a single operation.

If $X^1$, $X^2$, and $X^3$ are spliced together directly after the fully connected layer, the number of parameters does not simply increase linearly but exponentially, which results in a long model operation time. Therefore, in this module, in order to reduce the number of parameters and improve the efficiency of information fusion, the design of the fully connected layer is adopted after splicing in stages, and the information of three different scales can be fused by adding a small number of parameters. After the module processing, it connects the shallow features with the deep features to increase the feature information in the decoding process, thus achieving the purpose of improving the segmentation accuracy.

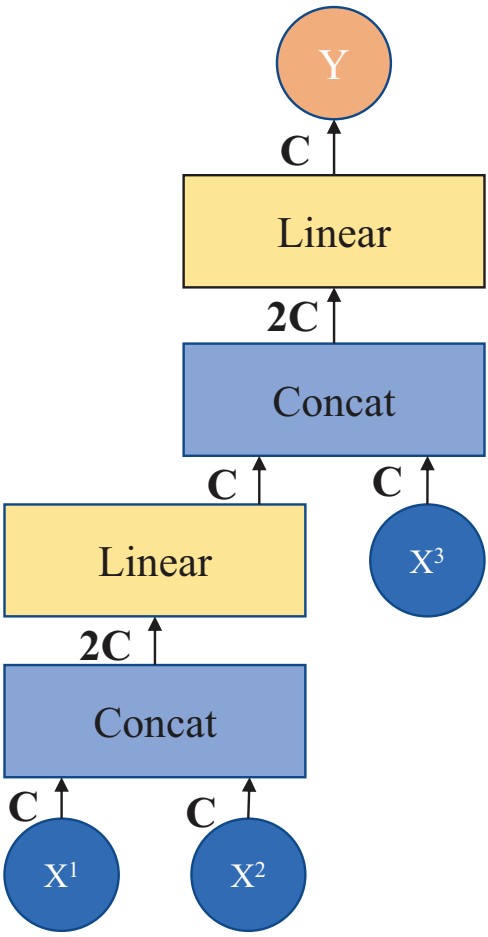

**Figure 3 Patch splicing layer.**

## Skip connection

The skip connection plays a key role in the U-shaped segmentation network by combining shallow, low-level, fine-grained feature maps from the encoder sub-network with deep, semantic, coarse-grained feature maps from the decoder sub-network. Connecting the different features through skip connections reduces the loss of spatial information due to downsampling.

## EXPERIMENTS

### Datasets

**"Chest Xray Masks and Labels" dataset**: This dataset (*Jaeger et al., 2014*; *Candemir et al., 2014*) contains the X-ray masks of chest and the corresponding labels; there are 704 images divided as training set and six images divided as test set. The average Dice Similarity Coefficient (DSC) and average Hausdorff Distance (HD) is used as evaluation metric to evaluate our model for lung segmentation in chest.

**Table 1 Comparison on the "Chest Xray Masks and Labels" dataset (average dice score % and average hausdorff distance in mm, and dice score % for each organ).**

| Framework | | Average | |
| --- | --- | --- | --- |
| Encoder | Decoder | (DSC)↑ | (HD)↓ |
| R50 | U-Net[a] | 97.43 | – |
| CNN | FCN[b] | 97.66 | – |
| R50 | Deeplab-V3[c] | 97.75 | – |
| R50-Vit | TransUNet[d] | 97.76 | 4.77 |
| Swin-Transformer | SwinU-net[e] | 97.23 | 4.53 |
| Our model | | 97.86 | 4.37 |

**Notes:**
[a] U-Net (*Xiao et al., 2018*).
[b] FCN (*Long, Shelhamer & Darrell, 2015*).
[c] Deeplab-V3 (*Chen et al., 2017*).
[d] TransUnet (*Chen et al., 2021*).
[e] SWinU-net (*Cao et al., 2021*).
DSC, Dice Similarity Coefficient; HD, Hausdorff Distance.

## Implementation datails

The model was implemented based on Python 3.9.7 and PyTorch 1.11.0. For all training image cases, data augmentation was used to increase data diversity. The input image size is set to 224 × 224, and the patch size is set to 4. We train the model on a NVIDIA Geforce RTX 3060 Laptop GPU with 6 GB memory. The SGD optimizer with momentum 0.9 and weight decay 1e−4 settings is applied to optimize the regression propagation of our model. Due to the small number of images in the medical image dataset and the unavailability of pre-training on a large dataset, the swin-tiny-patch4-window7-224 weights from Swin Transformer are introduced into the network for subsequent training using Transfer learning.

## Experiment results on Chest X-ray Masks and labels dataset

The segmentation results using different networks on the "Chest Xray Masks and Labels" test set are shown in Table 1. Our optimized algorithm achieves 97.86% performance on the DSC evaluation index. Compared with U-Net based on CNN neural networks, TransU-Net combined with CNN network and Transformer, the accuracy of SwinU-net before optimization is 0.43%, 0.1%, 0.63%. That is to say, our method achieves a better segmentation prediction effect. After comparison, it can be proved that the design with the special fusion module added by our design helps to improve the accuracy. The method of two fusions from the encoding process can better learn the global and long-distance semantic interaction information so as to achieve a better split effect.

The segmented images automatically output through the network can visualize the shape of the lung and its position in the chest cavity, as shown in Fig. 4, which can assist doctors in the diagnosis of lung defects and greatly improve the efficiency and accuracy of diagnosis.

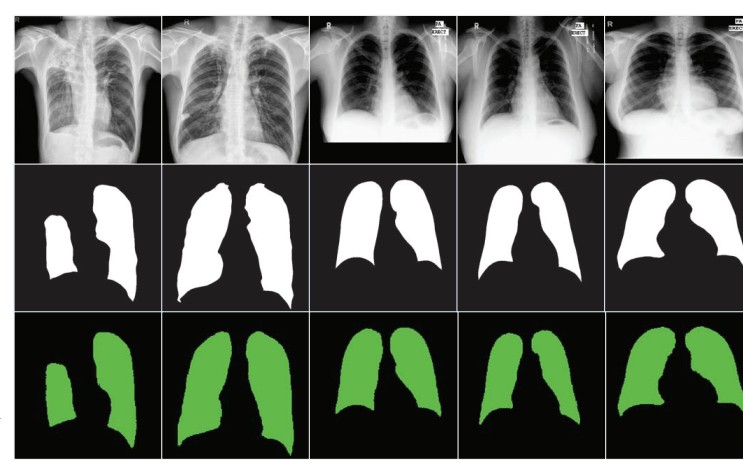

**Figure 4** **The segmentation results of the optimized model on the "Chest Xray Masks and Labels"** **dataset.** Image credit: "Chest Xray Masks and Labels" dataset (https://www.kaggle.com/datasets/nikhilpandey360/chest-xray-masks-and-labels); License: CC0: Public Domain.

**Table 2** **Comparison on COVID-19 CT scan lesion segmentation dataset (average dice score % and** **average hausdorff distance in mm, and dice score % for each organ).**

| Framework | | Average | |
|---|---|---|---|
| **Encoder** | **Decoder** | **DSC↑** | **HD↓** |
| R50-Vit | TransUNet | 85.50 | 16.53 |
| Swin-Transformer | SwinU-net | 82.18 | 20.71 |
| Our model | | 86.34 | 13.75 |

**Note:**
   DSC, Dice Similarity Coefficient; HD, Hausdorff Distance.

## Experiment results on COVID-19 CT scan lesion segmentation dataset

Due to the small number of samples in the "Chest Xray Masks and Labels" dataset, training was performed in the COVID-19 CT scan lesion segmentation dataset as a supplement to perform medical image segmentation. The dataset contains 2,729 samples, and a 9:1 ratio was used to divide the training and validation sets. The results in Table 2 show that our network still achieves excellent performance with an accuracy of 86.34%, which also indicates the good generalization ability and robustness of our method. Our network can perfectly perform the segmentation task in the irregular and complex COVID-19 CT and obtain suitable segmented images for review and identification by professionals.

## Ablation experiments on the "Chest Xray Masks and Labels" dataset

Because the data set authors set too few test samples, the test error may be too large. Therefore, the data set was adjusted, and the samples in the original training set were re-divided according to the ratio of 9:1 for the next stage of the ablation experiment.

From the results of the ablation experiments in Table 3, it can be concluded that adding skip connections can help improve the accuracy, and using the special splicing module we built can slightly improve the segmentation accuracy, but using the special splicing can

**Table 3 Ablation experiments on the "Chest Xray Masks and Labels" dataset (average dice score % for each organ).** Different conditions were set for comparison experiments, and the middle parameter was the average Dice Similarity Coefficient (DSC) results of training.

| Framework | Patch splicing | No patch splicing |
|---|---|---|
| Add 1/4 connection | 96.18 | 96.14 |
| Add 1/8 connection | 96.24 | 96.21 |
| SwinU-net | – | 95.93 |
| Add 1/4+1/8 connection | 97.37 | 97.31 |

reduce model parameters. Due to the full connection operation used when splicing skip-connected data, the number of direct splicing parameters increases exponentially, so we use two-stage full connection operations to achieve the same effect as the original direct splicing while reducing parameters.

## CONCLUSIONS

Our optimized pure Transformer encoder-decoder network can automatically segment lung parenchyma from chest X-ray images. Use the Swin Transformer block as a feature extractor to extract feature information, and use skip connections and our special splicing to learn long-distance semantic information interactively.

One of the more advanced methods at this stage is the combination of CNN and Transformer, such as TransU-net, and the other is a U-shaped segmentation network composed of pure Transformer, such as SwinU-net. The former category combines the advantages of CNN and Transformer to complete the task well, but for the small number of samples in the medical data set, the generalization ability is not as good as the network composed of pure Transformer like in this article. The pure Transformer model has the disadvantage of being insensitive to local perception, but we use migration learning to use module weights trained on large-scale datasets and use skip connections and splicing fusion to improve long-distance information interaction and global modeling capabilities, making up for it. shortcoming. The final experiments show that our model has good generalization ability and excellent segmentation effects.

However, our network can only segment 2D images, and there is a need for stereoscopic segmentation of 3D medical images. Therefore, the next stage of segmentation and application of 3D medical images is our goal and direction.

## ACKNOWLEDGEMENTS

We thank the publicly available datasets from National Library of Medicine, National Institutes of Health, Bethesda, MD, USA, and Shenzhen No. 3 People's Hospital, Guangdong Medical College, Shenzhen, China, for our research work.

### Funding

The authors received no funding for this work.

## Competing Interests

The authors declare that they have no competing interests.

## Author Contributions

- Yongping Dan conceived and designed the experiments, performed the experiments, analyzed the data, performed the computation work, prepared figures and/or tables, authored or reviewed drafts of the article, and approved the final draft.
- Weishou Jin conceived and designed the experiments, performed the experiments, analyzed the data, performed the computation work, prepared figures and/or tables, authored or reviewed drafts of the article, and approved the final draft.
- Zhida Wang performed the experiments, prepared figures and/or tables, and approved the final draft.
- Changhao Sun performed the experiments, prepared figures and/or tables, and approved the final draft.

## Data Availability

The code is in Supplemental File and Zenodo:

Weishou. (2023). Code and Data [Data set]. Zenodo. https://doi.org/10.5281/zenodo.7997332

Additional data is available at Kaggle:

Database name: Chest Xray Masks and Labels Pulmonary Chest X-Ray Defect Detection

Owner: Nikhil Pandey

URL: https://www.kaggle.com/datasets/nikhilpandey360/chest-xray-masks-and-labels

COVID-19 CT scan lesion segmentation dataset

Owner: Maede Maftouni

URL: https://www.kaggle.com/datasets/maedemaftouni/covid19-ct-scan-lesion-segmentation-dataset.

## Supplemental Information

Supplemental information for this article can be found online at http://dx.doi.org/10.7717/peerj-cs.1515#supplemental-information.

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
