# Peer review of "Optimization of U-shaped pure transformer medical image segmentation network"

_PeerJ Computer Science, doi:10.7717/peerj-cs.1515_

## Round 0.1 · original submission · Major Revisions

The reviewers have substantial concerns about this manuscript. The authors should provide point-to-point responses to address all the concerns and provide a revised manuscript with the revised parts being marked in different color.

Reviewer 1 ·

Basic reporting

The authors propose a new approach that utilizes deep neural networks based on U-shaped structures to assist in positioning and observing the shape of lung images. Specifically, they introduce an optimized pure Transformer U-shaped segmentation network that achieves better segmentation accuracy than previous methods.

Strengths of this study include its focus on a critical area of medical imaging research, the use of advanced deep learning techniques, and the development of a novel approach to improve segmentation accuracy.

Experimental design

The Results section should be presented as a separate section. The current number of experiments performed in the paper is not sufficient, and the author should consider conducting ablation experiments and including additional evaluation methods to demonstrate the effectiveness of the proposed method.

Consider moving Figure 4 to the Results section to better align with the presentation of the results.

Validity of the findings

Although the paper includes skip connections and special splicing processing in the normal segmentation method, it lacks ablation experiments to prove that these two parts can reduce information loss and increase model performance. Consider conducting these experiments to further support the validity of your approach.

Cite this review as

Reviewer 2 ·

Basic reporting

This paper proposes an optimized pure Transformer U-shaped segmentation network for medical image segmentation tasks, specifically for lung diseases. By adding skip connections and special splicing processing, the network achieves 97.86% accuracy in segmenting Chest Xray datasets, surpassing traditional convolutional networks and Transformer-convolutional network combinations.
Limitation
1. The paper would benefit from improvement in its language and grammar. There are several instances of small mistakes that need correction, such as the incorrect use of "As shown in the figure" in line 213. Consider revising the language to ensure clarity and precision throughout the paper.

Experimental design

2. It is important to provide more detailed information about the dataset used in the study. Specifically, please clarify the criteria used to divide the 704 images into the training set versus the 6 images used in the test set. Given the small size of the test set, it is important to demonstrate that your results are robust and generalize well.

Validity of the findings

3. The paper would benefit from a discussion section that provides insights into the results obtained. Specifically, please discuss why the proposed method performs better than other methods, whether there are other research studies that have investigated similar problems, and what the advantages and disadvantages of the proposed method are compared to state-of-the-art methods.

Additional comments

4. Lastly, annotate the full name of DSG and HG in Table 1.

Cite this review as

Reviewer 3 ·

Basic reporting

More background is needed for the application of segmentation in medical images, current performance, and problems. There is a lot of background in computational methods but not enough for the application perspective.

Is the abbreviation in Table 1, DSG and HG, actually DSC and HD? If not, please define them first.

Experimental design

The test set seems small. The authors also need to compare methods on multiple datasets (medical-related or general ones), including harder ones. The model performance seems to be similarly good in Table 1. Can the author find application examples that their method significantly improves?

What part of the images contributes to the final segmentation? Can the authors give feature importance plots and see whether that makes sense?

Validity of the findings

no comment

Cite this review as

---

## Round 0.2 · Minor Revisions

The authors addressed all major concerns. I recommend accepting this manuscript after the authors addressed a minor concern to clarify "the adjusted dataset" in Line 231 mentioned by Reviewer 1, as well as some other typos throughout the manuscript. A proofread is required.

Reviewer 1 ·

Basic reporting

The authors have addressed most of my concerns after revise. Line 231 "The dataset was adjusted", please clarify "adjusted". Is this referred to training ratio or any augmentation?

Experimental design

The authors have addressed most of my concerns after revise

Validity of the findings

The authors have addressed most of my concerns after revise

Cite this review as

Reviewer 2 ·

Basic reporting

Clear and professional English is used throughout.
Literature references are provided well.

Experimental design

The modified version satisfied the experimental design well.

Validity of the findings

The findings are validated well.

Additional comments

I agree to publish this paper.

Cite this review as

---

## Round 0.3 · accepted · Accept

The authors have addressed all of the reviewers' comments and I suggest accepting this manuscript.